# A Qualitative Investigation of Factors Affecting Saudi Patients’ Communication Experience with Non-Saudi Physicians in Saudi Arabia

**DOI:** 10.3390/healthcare11010118

**Published:** 2022-12-30

**Authors:** Mohammad Alkhamees, Jackie Lea, Md Shahidul Islam, Ibrahim Alasqah, Haitham Alzghaibi, Mohammad F. Alharbi, Fahad Albejaidi, Yasir Hayat Mughal, Vicki Parker

**Affiliations:** 1Department of Health Administration, College of Public Health & Health Informatics, Qassim University, Albukayriyah 52741, Saudi Arabia; 2School of Nursing and Midwifery, University of Southern Queensland, Toowoomba, QLD 4350, Australia; 3School of Health, University of New England, Armidale, NSW 2351, Australia; 4Department of Public Health, College of Public Health and Health Informatics, Qassim University, Albukayriyah 52741, Saudi Arabia; 5Department of Health Informatics, College of Public Health and Health Informatics, Qassim University, Albukayriyah 52741, Saudi Arabia

**Keywords:** patients, Saudi Arabia, thematic analysis, interviews, communications, culture

## Abstract

(1) Background: Communication is the main issue between the patient and physician. Communication gaps lead to medical errors, mistakes, and wrong diagnoses and treatments. It would certainly be a cause of patient dissatisfaction, the physicians’ job dissatisfaction, and the increase in the cost of health services. (2) Objectives: The objective of the study was to explore the perception of Saudi patients when they meet non-Saudi consultants at hospitals. (3) Material and Methods: This study is exploratory in nature. Semi-structured face to face interviews were conducted. Study participants were patients from the outpatient department of the Hospitals of Qassim province, Saudi Arabia. A total of eight respondents, four male and four female, participated in the study. (4) Results: Interviews were transcribed verbatim and analyzed using a thematic analysis approach. Two main themes emerged: the first theme is encountering difference, with three sub-themes, while the second one is the absence of partnering, with four sub-themes. (5) Conclusions: The analysis of the patients’ experiences of their medical encounters revealed that the effectiveness of the physician-patient communication was jeopardized by two challenges: encountering differences and the absence of partnering. (6) Limitations: This study was conducted in one site only and hence findings must be tested in application to other sites in other parts of Saudi Arabia and in other similar countries. A further limitation of this study was the cultural barrier that was encountered by the researcher during the interview process.

## 1. Introduction

Saudi Arabia’s healthcare systems have vastly improved over the last twenty years, and there is an unprecedented demand for health services. Moreover, this demand is expected to increase for decades to come, mainly due to a rapidly growing and ageing population, and a sharp increase in chronic diseases [1]. Saudi Arabia not only has the highest population growth rate in the world at around 3% but has experienced a huge increase in non-communicable diseases (NCDs), such as diabetes and hypertension [2], which additionally contributes to the increasing demand for health services. At present, there are insufficient Saudi national physicians to deliver healthcare to the Saudi population [1]. As a result, foreign medical professionals are used to fill the gap between the increased demand for health service and the shortage of Saudi physicians. This scenario has not improved due to the shortage of trained local physicians and insufficient Saudi medical graduates [3]. According to the most recent Saudi Ministry of Health statistics, non-Saudi physicians comprise 78% of the total national physician workforce, and national medical schools are unable to produce enough graduates to match the demand for healthcare services [4]. The number of foreign physicians employed has increased from 6215 in 2015 to 6455 in 2016 [5]. Two out of every three medical consultations are likely to occur with a non-Saudi physician [1]. Thus, the need to ascertain the impact of cultural and language differences between healthcare professionals and patients in terms of the quality of communication and delivery of healthcare services is clear. Communication is held responsible for providing high quality services. The communication gap between the physician and patient is found to be responsible for fifty percent of medical errors and mistakes, wrong diagnoses, and treatments [6,7]. Poor communication would lead to the physicians’ job dissatisfaction, patient dissatisfaction, and an increase in health costs [8]. Culture plays a significant role in effective communication between patients and physicians [9,10]. The perspective of health culture has an influence on the patients’ values, norms, traditions, and beliefs, as well as when expressing symptoms, the patients can easily inform their physicians what they expect from him or her. In Saudi Arabia the majority of health professionals are non-Saudi. Due to these issues, Saudi patients are having difficulties in communication with their non-Saudi physicians [11,12]. Due to this unique setting in healthcare organizations (HCOs), there is an immense need to highlight this issue and provide solutions for this issue. As discussed above, the majority of the physicians are expatriates and it is likely possible that those patients encounter non Saudi physicians at some point. An increase in demand for health services and due to less availability of Saudi physicians, this problem will likely exist for the coming years. This is one of the pioneer studies that would shed light on patient-physician communication and would suggest some strategies to obtain optimal health results. This study got support from the theories of social and cultural reproduction and social relation introduced by Bourdieu [13]. Three concepts were introduced in the theories of Bourdieu which have been widely recognized by researchers and scholars. These are habitus, capital (cultural capital), and field to investigate the patient-physician communication in an intercultural setting. Habitus explained the influence on behavior and the perception of an individual and let us know that the patients’ culture cannot be separated from their health issue. Habitus is also influenced by the socialization process. Habitus enables an individual to deal with day-to-day unseen and changing events; basically it helps to create principles so that one can handle the unforeseen events effectively [14]. One of the key elements embedded in the habitus nature is the way individuals meet, speak, talk, and greet each other, and language use and the accent with which individuals speak; all of which they do without consciously thinking. Likewise, capital, the second most important concept in social and cultural reproduction theory, sheds light on the physicians’ assets and qualities. Bourdieu refers to it as linguistic capital. A lack of knowledge by the physician, and a lack of exposure to the patients’ culture diminished the cultural capital and, likewise, the lack of medical knowledge by the patient, compromising the ability of the patient to communicate effectively with physician, is called institutional capital [15]. High cultural capital gives more power to an individual/agent in the society in a relevant field. Cultural capital is more valuable if it indicates something more worthy in society in a specific area. This implies that capital empowers its holder, dictates one’s position, and forms the base of the social structure within the society’s’ hierarchy. On the other hand, capital has been criticized in all of its forms for inequality because those who hold less power and capital are considered as less privileged in society. Cultural capital is further divided into embodied, objectified, and institutionalized forms [16]. The third important concept is field: the place where communication takes place between the physician and patient. In field, interactions between the patient and physician occurs. One cannot limit the field to a place or institution; rather it can be any form such as a school, hospital, family, profession, market, town, or it might be a series of autonomous but structurally homologous fields [17]. The aim of this study is to explore the experience and perceptions of Saudi patients with non-Saudi physicians in HCOs during medical consultations. This paper tries to answer the following research question: 

RQ1: What are the communication experiences of Saudi patients with non-Saudi physicians during medical consultations? 

## 2. Materials and Methods

### 2.1. Methodology/Study Design

The study is exploratory in nature, and an inductive approach was adopted. Semi-structured in-depth interviews with patients were conducted over a six-week period. Each set of interviews was collected and analyzed independently. However, interviews with patients were conducted concurrently to efficiently use the time designated for the data collection and, most importantly, to meet with participants at their available time and date. A specialist hospital was selected to recruit the study participants (Table 1).

### 2.2. Description of the Research Context

The specialist hospital was chosen because it is one of the largest hospitals in the state of Qassim, so recruiting sufficient numbers of participants would be achievable (Table 2). Secondly, this institution not only has adequate outpatient clinics, but they take place in an independent building, separate from the main hospital building. This benefited the study in terms of recruitment and access. Public hospitals can be a true representation of a population with diverse characteristics. The researcher identified the Qassim region, particularly Buraydah city, due to its suitability for assisting to meet the study purpose. Buraydah is the largest city in the Qassim region and the hometown of the student researcher. Thus, a convenience sampling strategy was applied to justify Buraydah city as the study location. Important in the context of this study, Qassim has one of the largest contingents of foreigner physicians, comprising almost 85% of the physician workforce [4].

#### 2.2.1. Data Collection Technique

Semi-structured interviews were conducted of the study participants. The key question asked in the interviews was “What are the inter-cultural communication experiences of Saudi patients with non-Saudi physicians during medical consultations?” Two main themes emerged; the first theme is encountering difference, with three sub-themes, while the second one is the absence of partnering, with four sub-themes (Table 3). The data collection was conducted from June 2016 to September 2016. There were a number of initial challenges during the data collection stage. The researcher decided to use flyers as the main recruitment strategy for patient participants. Many studies have applied this method in order to recruit participants [8]. In this recruitment strategy, potential participants read the flyers and, if interested, identify them, and ask to participate in the study. Self-selection is considered the most ethical approach in participant recruitment [17]. To recruit potential patient participants, the researchers distributed the study flyers in a suitable location within the outpatient ward. The researchers contacted the interested participants through a third party (gatekeepers). In this study, the receptionists of out-patient clinics acted as gatekeepers. Gatekeepers are very important facilitators in recruiting participants; thus, the researchers explained the purpose of the study to them and worked at building a good relationship with them [18].

The following main themes and sub-themes have emerged from the analysis of the transcriptions.

#### 2.2.2. Data Analysis Techniques

All interviews were transcribed verbatim. In the current study, a thematic analysis approach using Braun and Clarke’s [19] six step model for qualitative data analysis and interpretation techniques was used. The guidelines of the thematic analysis by Braun and Clarke [19] proposed that the researcher first familiarize him/her self with the data collected, followed by generating codes, searching for themes, doing thematic review, defining and naming themes, and finally writing a report. The familiarization with the collected data begins when researcher started interacting with the interviewees to conduct interviews. The researcher deeply involved him/herself in transcribing the interviews. Transcription is the key stage of the data analysis, because, in this stage, final themes have to emerge. After the transcription, the researcher immersed him/herself to write initial notes, thoughts, and memos. The aim of this stage was to build a strong general sense of the complete picture about the communication experience of the study participants, rather than reaching a premature conclusion. All relevant sentences and phrases were highlighted and coded, and similar codes were grouped. Each code was given a descriptive title to capture the nature of its contents.

The coding process started with labelling (naming) the data extracts with specific labels, rather than general ones, to reflect each participant’s experience precisely [20]. In addition to coding manually, the researcher used the qualitative data analysis software (NVIVO) application to code the data after uploading MS Word transcripts of all of the participants into NVivo. This was effective in enhancing the management of the vast amount of data that the interviews produced. Translating the codes into themes occurred when the researcher re-focused the analysis at a broader level. A theme is not only an outcome of coding and categorization, but it is also an analytical reflection on the data by the analyst [20]. Refining and defining began after reaching a satisfactory thematic map reflecting the collected data. The process of refining and defining involved identifying the essence of each theme, as well as the overall of themes. The activity of producing the final report began after establishing a set of fully worked-themes.

##### Ethical Consideration

The research was approved by the Human Research Ethics Committee of both the University of New England (HE16-079) and the Ministry of Health in Saudi Arabia (H-04-Q-001). In addition, the researcher completed a National Institutes of Health (NIH) training course on protecting human research participants. The key ethical issues in conducting the study included concealing the participants’ identity, providing appropriate information to ensure that free and voluntary consent to participate was obtained, and that participants were not coerced to participate. In addition, the researcher had to take measures to ensure that the translated data was accurate and a true translation of the transcripts. In terms of the participants’ identity, all participants were assured that their identities would be concealed from the public. Participants were allocated a pseudonym so that their privacy would be protected when reporting the study data [21]. The consent form and information sheet for participants were provided in both English and Arabic. All potential participants were given a copy of the information sheet for participants prior to agreeing to participate. Just prior to the commencement of the interview, participants were given time to reread the latter document which included information about the researcher, the study, and the participation requirements. Prior to the commencement of every interview, the researchers presented the consent form to the participants. All participants signed the consent form willingly. All interviews were conducted in the hospital in a quiet private room that was provided by the hospital administration for the researcher’s use during the data collection period.

## 3. Results

### 3.1. Theme One: Encountering Difference

This theme is about the essential role that difference plays in patient and physician encounters, and its impact on patients’ consultation experiences and their perceived health outcomes. This notion of difference is both preconceived in the patients’ mind that sees non-Saudi physicians as ‘other’ and constructed during the medical encounter with the influence of cultural differences, such as language, religion, and customs. Patients felt unable to relate to non-Saudi physicians and experienced a communication disconnection during the medical encounter. However, on a few occasions, patients felt that their culture was acknowledged and integrated in their medical treatment, which ultimately enhanced communication and patient satisfaction. This theme encompasses three sub-themes: being different, being disconnected, and being culturally acknowledged.

#### 3.1.1. Sub-Theme One: Being Different

This sub-theme highlights the ways in which patients perceive non-Saudi physicians as being different and unlike them. This perception of difference was reflected throughout the patients’ narratives. It underpinned the patients’ anticipation that they would have difficulty communicating and developing a mutual understanding with non-Saudi physicians, as the following quote illustrates: 


*“I believe that communication between Saudi patient and non-Saudi doctor is difficult by nature because of the different cultures. It would be much easier to communicate with a Saudi doctor. Saudi doctors understand us better because we have the same country, culture, and language”*
(Participant 1).

Participant 1 came to the medical encounter with a pre-existing belief that the differences would make communication with non-Saudi physicians difficult. She recognizes differences as natural, and believes she cannot do anything to change this. This natural difference is manifested through ethnicity, language, religion, and culture. For this reason, she prefers to communicate with Saudi physicians and believes this to be the only way for a better understanding. She said: 


*“There will be no communication if the patient and the doctor share no common language”*
(Participant 1).

Participant 2 sees the difference as evident, particularly in relation to religion as she said:


*“On some occasion, they came to give my mam the back injection but they found her bed positioned toward Mecca because she was performing the prayer. Even though, they left and came back later, I felt some of them did not like this delay at all. Some of them are non-Muslim”*
(Participant 2).

The above example indicates that Participant 2 sensed that physicians were not comfortable with the delay caused by the patient performing prayers. Thus, Participant 2 saw the difference in religion with a non-Saudi physician as an obstacle that prevented physicians from recognizing and understanding what is important for Saudi patients. Unlike the above participants, Participant 7believes that differences with non-Saudi physicians go far beyond language and religion and are all encompassing.

She stated: 


*“Of course, they seem different from us on everything”*
(Participant 7).

The above quotes illustrate how patients have characterized their perceived differences with non-Saudi physicians, which implies a pre-existing belief and an expectation of difference in the patient’s mind, even before meeting any physician, leading them to anticipate difficulties, and thus influencing their preparedness to communicate their health concerns. Evidence of categorizing non-Saudi physicians as ‘other’ was strongly expressed in all participants’ stories, with the common reference to ‘us’ and ‘them’. Moreover, the patients’ strong sense of being different from non-Saudi physicians can be inferred from the way in which patients have referred to non-Saudi physicians as ‘non-Muslim’, ‘foreigners’, and ‘strangers’, and constantly comparing them to Saudi physicians.

Participant 6 asserted: 


*“There is no way that the doctor will understand you unless the doctor is a Saudi. Though they were not asked about their medical experiences with Saudi physicians, patients talked about them in comparison to their experiences with non-Saudi physicians to show their preferences, and to highlight the differences and difficulties”.*


This is exemplified by Participant 3’scomment:


*“I would prefer a Saudi physician who can exchange information with you and can understand you right away. I can communicate with Saudi physicians by informal language. We use informal language in our daily life so it would be easier if you could explain your health issues to your physician using informal speech. As you know, the only way to do that is to have a Saudi physician. I prefer to see Saudi physicians because they are one of us and they can understand us better than foreigner doctors”*
(Participant 3).

From Participant 3’sperspective, even though she met with a physician who speaks the same language, she believes that Saudi physicians will understand her better because it is more than the words that are spoken, but how they are spoken in terms of formal or informal language. Thus, Participant 3 and other participants indicated a preference to be seen by Saudi physicians to whom they feel alike and related. 

Participant 3 refers to Saudi physicians as ‘one of us’, and non-Saudi physicians as ‘strangers’. This sub-theme has shown how patients perceive non-Saudi physicians. It has demonstrated that patients’ pre-existing conceptions of non-Saudi physicians as different and ‘other’, have negatively influenced their expectations regarding communication and understanding when meeting physicians. As a result of this expectation, patients are less likely to engage with the physician in the medical consultation, which in turn disconnects the two parties.

#### 3.1.2. Sub-Theme Two: Being Disconnected

This sub-theme describes the patients’ experience in regard to their ability to connect with their physicians. Patients were challenged and, on some occasions, experienced a total disconnection with their non-Saudi physicians because of differences in language, religion, and customs. There is a sense of frustration and concern among the patients. They were frustrated due to not being able to have successful communication and were concerned about how this would reflecton their health problems.

A major challenge experienced by patients was not being able to understand and to be understood by their physicians, as Participant 4 describes:


*“I don’t really know if the doctor got and understood what I have already told him. This happens with doctors speaking with broken language or don’t speak Arabic at all. I use non-verbal language to explain to doctors my health problem. It is not sufficient for him and for me. We as patients suffer a lot from seeing non-Arabic doctors. For example, I can convey to the doctor that my stomach hurts me but I cannot tell him any other information such as which part of my stomach is the source of pain, or the food that I had in which I think is the cause of my stomach-ache. Believe me, it is hard to communicate with doctors who speak no Arabic”*
(Participant 4)

Participant 4 had to resort to communicating with his physician non-verbally in order to overcome the language barrier. He was frustrated and went through unnecessary suffering and hardship because he was not sure if the physician had comprehended his health complaint. He believes that this type of communication does not satisfy even the minimum consultation standard, in which a physician has to hear the symptoms and the exact location of the pain in order to provide appropriate treatment. Participant 4’sexperiences illustrate how the absence of a common language limits effective communication and information sharing that is important for diagnosing illness and ensuring the patient safety. The absence of a common language restricted the patients’ ability to fully describe their problems to their physicians, which created significant distress for them. Unlike the rest of the participants, Participant 8 regarded the language difference as less of an issue because patients use the help of whomever accompanies them, such as friends, relatives, or often nurses to translate. He said:


*“Language is not a problem because always there is someone to do the translation such as the nurse. However, the majority of nurses are not Arabic speakers, only a few of them 95 speak very limited Arabic”*
(Participant 4)

As Participant 3 explained:


*“Most nurses here are from non-Arabic speaking countries such as Philippines, and India. So, they know only few Arabic words such as go, come, visit, and so on. It seemed from the patients’ experiences that nurses could help with translation but not enough to replace the need for a qualified translator”*
(Participant 3).

Moreover, Participant 5believesthat, even with the provision of a translator, communication problems are not completely eliminated. He said:


*“I think that language issue is important. We could use a translator between the doctors and patients who do not speak the same language. Even though, I prefer the contact between the patient and the doctor directly because sometimes the translators do not convey the information accurately between the two parties”*
(Participant 5).

This quote shows that Participant 5 struggled during the medical encounter to understand and to be understood by his physician. Even with the presence of a translator, language differences cannot be completely eliminated, as Participant 5 suggested, because a health message loses some of its content and effectiveness when translated by a third party. Even a skilful interpreter cannot translate the patient’s exact emotions and feelings. Patients have different views withregard to the use of a translator. For some, the presence of another person, such as a translator, would increase the stress rather than reduceit, as was the case for Participant 1 who said:


*“In some medical visit I got reluctant to talk to my doctor because another person will be in the room like another doctor or nurse”.*


Participant 2 “*insisted that it is not enough to have a translator, but the translator has to be Saudi: Having a local translator is important to use simple words and informal language. Patients found that language was an obstacle not only with physicians who do not speak Arabic, but at times also with physicians whose native language is Arabic. Even though Arabic physicians and patients speak the same language, it is spoken differently in Saudi Arabia than in the physician’s own country and is compounded by the existence of a local dialect with its own unique features*”.

For example, Participant 6 said: 


*“if you tell an Arabic doctor that I have “mukuse” [abdominal colic] there is no way that the doctor will understand you. The following quote by Participant 3 clearly confirms Participant 6concerns and exemplifies how the same language can be used differently by different people. Even though some non-Saudi physicians speak Arabic, I find difficulties to make some of them understand me because we use words that they do not know what it means. For example, in one of my medical visits I told the doctor that I have tanmil (numbness) in my left hand. The doctor did not understand what the word tanmil means because this word is used by Saudi as it seemed to me. In the third visit, he came to know what the word or the phrase means. I got myself embarrassed because I was not able to explain it to him. If he had understood me from the beginning, he would have given me the right medicine in the first place because I felt better when he changed my prescription. I suffered for quite some time because he did not understand what the word tanmil is. If he had understood me right in the first place, he would diagnose my illness accurately and prescribe the right medicine for me”*
(Participant 3).

Participant 3’sdistress began when she used a local word that seemed to be sufficient forher in describing her unbearable pain. This word was important because it was meant to help the doctor diagnose her illness. However, this local word was not known or understood by the physician. Therefore, Participant 3 was emotionally stressed and frustrated because she could not replace it with another word that has a shared meaning for both of them. In addition to that, she experienced physical suffering because her health problem was not treated until the third visit when the doctor finally came to understand what she meant by the word (tanmil). In short, Participant 3 experienced unnecessary emotional stress and physical hurt and her safety was jeopardized because she was given the wrong prescription. Even though language is a major issue for Saudi patients, it was not the only cultural barrier that made Saudi patients feel distanced and disconnected from their physicians. Religion and custom barriers were very much present during the patients’ medical consultations. Saudi patients are very attached to their religion and customs, so they expect to be treated accordingly by their physicians. However, on some occasions, patients felt that their attending physician was either not aware of their cultural needs or not willing to take them into consideration. For instance, Participant 2 left the medical consultation because she felt that her religion and custom needs were trivialized. She said:


*“Anyway, let me tell you about my experience with my previous dentist. During the medical visit, I will uncover not my whole face but to a point that allows the doctor to work on my teeth. He did not like me doing that and asked me to take the whole veil off. He told me that his intention is to finish the job perfectly not to stare at my face. I left him and moved to another doctor”*
(Participant 2).

From Participant 2’s perspective, culture should not be separated from her health problem. She expected the dentist to fix her dental problem without compromising any of her cultural values, particularly the ones that are related to gender norms and modesty. The result of this medical encounter was that she was not happy and did not receive treatment. Additionally, the dentist was challenged inrecommending and applying the medical procedure or treatment he felt was necessary. This medical encounter might be classified as a ‘lose-lose situation’. When the patients’ cultural norms are different from those of the physician, establishing a patient-physician connection is very challenging. Moreover, patients may feel uncomfortable and stressed. That is exactly what happened to Participant 3, as she describes below:


*“The whole process of examining my neck was so embarrassing to me. Unlike the hand, the neck is a sensitive part of my body. Even though, the examination time is not too long, I felt so shy and I wanted the doctor to hurry up and finish. Moreover, the story did not end there because every time I come to see him, I have to uncover my neck to him. I do not understand the fact that he needs to check my neck in every visit. He diagnosed my illness in the first visit that should be enough. My suffering last for a year because I was schedule to see him in every month. Whenever he asks me to uncover my neck, I feel hurt like just he is asking me to show out my whole face. I carry such burden on the medical visit day and in my way to see him”*
(Participant 3).

Participants 2 and 3 decided to continue seeing the physician, which came at the expense of her emotional ease. Even though Participant 3 experienced a great deal of embarrassment and hurt, she remained with the same physician because she was desperate to get rid of her physical illness. Participant 3’s distress continued for a year, each visit causing her apprehension and stress because her cultural values were not considered in parallel with her physical illness. The physicians’ lack of recognition of her cultural values resulted in her feeling very uncomfortable and wishing for the encounter to finish quickly so she could leave. This sub-theme has illustrated how the cultural differences between the patient and physician can impede communication, disrupting, and possibly ending, the medical encounter. Cultural differences between patients and physicians impacted the sharing of information and the possibility of mutual understanding, the ability to reach an accurate diagnosis, as well as the patients’ perceptions of their safety. The following sub-theme illustrates how cultural differences can be used as a bridge for establishing communication. This was reflected in the participants’ positive experiences in which they felt that their cultural values and needs were acknowledged and integrated into their medical treatment.

#### 3.1.3. Sub-Theme Three: Being Culturally Acknowledged

This theme is about the centrality of cultural identity for patients. It describes how they reacted toward physicians who acknowledged them as a person whose identity and way of life is determined through their culture and religion. It is about how the cultural competence of the physician defines the nature of the patients’ communication experience. In this context, cultural competence means the physician’s awareness of and respect for the patients’ culture and his/her ability to express this when communicating with patients or providing them with treatment. Patients shared examples of some good communication experiences in which they felt that physicians were acknowledging them, recognizing their culture, and incorporating cultural considerations in their health treatment. This is best exemplified by Participant 2’s experience:


*“I do not like going to the dentist because I have to uncover my face, so he can work on my teeth. However, this doctor was very considerate because he allowed me to do whatever makes me comfortable in regard to taking my face cover off. So, I used to uncover only the part of the face that he needs to see in order for him to perform the job”*
(Participant 2).

Participant 2’s experience confirms the impact of a physician’s culturally respectful behavior in shaping and influencing the patient’s medical experience. Participant 2’s general attitude towards seeing a dentist was not favorable because she would usually be asked to uncover her face. She came to see the dentist expecting that she would have to compromise some of her cultural values, but to her pleasant surprise, she did not have to. This particular dentist was able to change Participant 2’sattitude towards seeing a dentist, at least for that particular visit. He was not only aware that cultural beliefs do not permit her to uncover her face before men who are not direct relatives, but he was also able to find a way to adapt the treatment to accommodate her beliefs and enhance her comfort. Consequently, Participant 2was engaged with her care and satisfied with the outcome. Similar to Participant 2, some participants recounted pleasant medical experiences in which they met physicians who acknowledged their culture.

Participant 6 was greeted and received in a customary Saudi way, which was sufficient to change his perception of his entire medical experience. He felt at ease, not only because of the physician’s initial welcome, but also because the reception he received from the physician was in accordance with Saudi custom. He said: the moment I entered the room, the doctor greeted me with a big smile and handshakes. In Saudi Arabian custom, a greeting and a handshake are a must when meeting someone. Participant 4’s experience was similar to Participant 6’s. Participant 4 said:


*“Some doctors are very welcoming and eager to start a discussion with you. For example, one time a doctor asked me about whether I went to see the traditional healer or not. I said no I didn’t go but if you do not heal me, then I will go. He laughed because of what I have said. He knew many things about the people in this city and their traditions. The funny thing was that this doctor was not even an Arabic doctor. He was from Russia or Greece”*
(Participant 4).

The physician’s awareness of Participant 4’sculture enabled a connection and helped to activate a conversation between the two parties. Participant 4 had welcomed the physician’s talk and was enthused about it. Participant 4’s appreciation of the physician’s knowledge about his culture can be sensed from his statement: He knows many things about the people in this city and their traditions. The physician’s knowledge and awareness of patients and their local practices bridged the cultural differences and made Participant 4 realize the commonalities they shared. Participant 4 experienced a feeling of relatedness due to the physician showing an interest in him and demonstrating an interest in the local culture. Participant 4 further admired the fact that the physician was knowledgeable about a culture so different from his own. Thus, a physicians’ knowledge of a patient’s culture can be used as a means to establish effective communication and build a relationship. Moreover, some patients experienced a situation where a physician allowed some of the patient’s commonly used traditional medicine practices to be integrated into the medical treatment. For instance, the use of honey for several types of illness is very common among Saudis as an alternative to modern medicine. This can be reflected in Participant 2 story below that shows a Saudi nurse proposing honey treatment:


*“I can tell you that one of the things that helped my mother’s wounds to heal is the use of alternative medicine. A Saudi nurse has some experiences in applying honey on the wounds to speed up the wounds to heal. With doctor approval and supervision, he continues to use the honey on the wounds. Frankly speaking, I and my family had witnessed huge improvements on my mom’s wounds”*
(Participant 2).

Even more satisfactory than the experiences of Participants 4 and 6, Participant 2 experienced both an acknowledgement of and the accommodation of her culture during the medical encounter. Thus, she and her family experienced a constructive medical encounter that resulted in the patient’s satisfaction and engagement, and a positive health outcome. In conclusion, the three sub-themes above have illustrated how the lack of shared understanding between Saudi patients and their non-Saudi physicians categorizes non-Saudi physicians as different and medical encounters with such physicians as alienating. Physicians being knowledgeable and familiar with the patient’s culture can reduce the disconnection that patients experience during the medical encounter. Without a shared cultural understanding and a willingness to compromise by the physicians and patients, the opportunity to develop a mutual relationship and to partner in healthcare decisions was also compromised.

### 3.2. Theme Two: Absence of Partnering

One of the most important reasons that prevented patients from establishing a rapport and effective communication with their physicians was their perception that they were not treated as a partner during the medical consultation. This theme highlights the lack of reciprocity and partnering in the patient-physician relationship that was reflected in the patient experiences, resulting in patients feeling powerless, overlooked, being unwilling to trust non-Saudi physicians, and perceiving the time given by their physicians as inadequate for meeting their needs and expectations. From the patients’ perspective, partnerships make a difference to the communication quality between the patient and physician. The level of patient satisfaction was directly related to the degree to which they were engaged and were enabled to participate in the medical consultation. The patient’s actual experiences were significantly different to what they had expected withregard to their engagement and participation in the medical consultation. They expected to be regarded as a partner, but their experience was otherwise. Thus, most patients viewed their medical consultations unfavorably. This theme encompasses four sub-themes; being powerless, being unnoticed and neglected, being reluctant to trust, and not being given adequate time.

#### 3.2.1. Sub-Theme One: Being Powerless

The most fundamental cause of the absence of a physician-patient partnership was the unequal distribution of power between patients and physicians. This was reflected in the patients’ feeling they had little control over what occurred during the consultation. Patients were disempowered from taking a lead in the conversation about their health, from participating in the conversation, and establishing a rapport with their physician. Their powerlessness was due to their beliefs about the roles played by physicians and patients, and due to the manner in which the organization’s culture, policy, and rules reduce their choices and participation. Patients came to the medical consultation with a belief and preconception that physicians are in charge and should decide on the course of the medical encounter, as the following quote illustrates:


*“I believe those doctors are more responsible about having good communication because patients do not have the same power as doctor. So, doctor should initiate asking questions and try to activate the patient into a conversation during the medical visit. Doctor must be the one who begin by exchanging information with the patient”*
(Participant 1).

Participant 1 came to the medical encounter already believing that physicians are in control. This was not raised as an objection, but it was her perception that it is normal for physicians to exert greater control than patients. She believed that physicians should be responsible for engaging the patients and initiating the exchange of information. Similar to Participant 1, Participant 4 believed that physicians are entitled to have more authority and control over the medical encounter than patients because of their knowledge. His beliefs limited his engagement and participation in the medical treatment and resulted in a passive acceptance of the physician’s thinking. He thinks that it would not be right to disagree with the physician who possesses medical knowledge, which he does not have.


*“I am a believer in specialization and this is his specialization. This is a doctor and this is his field. I am ignorant when it comes to health knowledge. So, why should I question what he thinks?”*
(Participant 4).

Unlike Participants 1 and 4 who are convinced that physicians should be in charge, Participant 6 has expressed his objection to the physician’s authoritarian attitude. He said: doctors must understand that patients should not be viewed as receivers but as partners. Some doctors think that they own the clinic. Participant 6’s quote indicates that he believes that physicians think that they are superior, so they do not perceive patients as partners. Along with feeling inferior in their communication with physicians, patients also described how hospital policies and rules have contributed to their powerlessness. This can be best exemplified by the following quote:


*“The other issue is that you can’t change your doctor because the hospital policy refuse this. I tried to change my doctor but I have been told by the hospital’s administrators that 104 either your doctor die or you leave the hospital otherwise you can’t change your doctor. I told them him that I have been with this doctor for 2 years and I feel no improvement in my condition but they refused to change my doctor”*
(Participant 4).

Participant 4’sexperience illustrates how the hospital’s policy can contribute to the patients’ feelings of having no control in managing their health, in particular, the option to consult with another physician if there is no evidence of health improvements, as in Participant 4’scase. Several female participants were disappointed by not being able to have a choice of physician. Most female participants indicated a preference to consult with female physicians. Participant 1 stated:


*“As a female patient, I would like to see a female doctor who can understand me better since we are alike. Failure to respond to patients’ preferences and choices made patients feel less in control and somewhat helpless to manage their own health issues, as well as disempowered to participate”*
(Participant 1).

This sub-theme has illustrated the prevalence of uneven power between the physician and patient. Patients have felt disempowered before, during, and after their medical encounters. This was attributed to the patient’s own belief and/or the organization’s policies. As a result, patients were not encouraged to take a lead in managing their health issues, or to establish a rapport with their physician. The lack of physician-patient partnership was also attributed to the patients feeling overlooked and neglected, as the next sub-theme will illustrate.

#### 3.2.2. Sub-Theme Two: Being Unnoticed and Neglected

This sub-theme highlights many patients’ feelings of being overlooked and disrespected, and the ways in which this undermined their sense of value and wellbeing. This sense of neglect was reflected throughout the patients’ stories. Participant 4 for example said: *You feel that they are neglecting you completely*.

Participant 7 described her experience of being neglected and how it made her feel:


*“When I went to see the doctor from Egypt, he did not bother to listen to me or give me a chance to explain my health problem and my concern. It is very important that doctors pay attention to their patients. I used to see a doctor and during the visit I get so nervous because he is not paying attention to me or to what I am saying. He talks to the nurse or gets himself distracted by anything like the computer while I am talking to him”*
(Participant 7).

Participant 7 felt totally disregarded by her physician. For her, paying attention goes beyond just hearing; it requires full concentration on what the other person is saying. She went to the hospital with the expectation of being given a chance to present her health problem to the right person: a sympathetic physician. However, she was disappointed by her reception and by the way she was ignored by her physician. Feelings of being overlooked, disrespected, and neglected were described by all participants, but each experienced it differently. Participant 6, for example, felt that he was not listened to and that he was rushed by his physician. He said:


*“He did not listen to me enough and he was kind of pushing me to cut off my talk.”*


Participant 2 felt disrespected by repeatedly being interrupted by her physician: They interrupt you frequently as a way to tell you to stop talking. Lena was not given a chance to speak. She stated: 


*“Some doctors do not give you a chance to talk. She described her unsatisfying interaction with a dermatologist. Another bad experience happened with me with a dermatologist. There was no exchange of information. I did not get what I wanted at all. She did not meet even the minimum of my expectations”*
(Participant 1).

The examples above indicate that patients found themselves left out of taking an active part in the medical consultation. The patients were unable to establish a relationship and form a partnership with their physicians. As a consequence, they were dissatisfied with their medical consultation.

From the perspective of patients, physicians failing to pay attention and provide patient-centred care was also attributed to the physicians’ tendency toward a biomedical approach. This was reflected in the patients’ descriptions of how they felt when physicians ignored them and focused more on medical assessment tools or medication. Patients understand that radiology and pathology tests are essential diagnostic methods, but they should not be used at the expense of human interaction. This can be best exemplified by Participant 8 quote:


*“Recently, I started to feel dizzy when I stand up. So, I came to see the doctor. The doctor didn’t bother talking to me to understand exactly what I have. He asked me to take some medical tests and come to see him a month later, can you believe it. Again, to what I have told you before, doctor concentrate always on medical test. Even after second visit, they still ask to do more medical test. For example, one doctor asked me to do transactional X-ray but I told him that I already have done that then he said ok then you need to do magnetic X-ray, can you believe that!”*
(Participant 8).

Participant 8 likes to be treated by his physician as a person who has a story to tell, as well as having views, values, and concerns. He felt emotionally hurt because the physician did not attempt to interact with him, giving him a chance to explain his health problem, or allowing him to state his opinions. Instead, the physician ignored him by shifting his entire attention to the medical test to diagnose his illness. He was incredulous about how he was treated and felt that the physician saw him as a health case rather than a person with a health problem. Participant 8 was not the only patient to experience a consultation characterized by a biomedical focus. Other patients felt that physicians are more focused on the medication than ministering and treating patients holistically. Participant 6 said:


*“I am asking for more attention and care from the doctor. You know some doctors just write a prescription. They do not give a patient enough time to explain his/her health problem”.*


Participant 6’s concern and complaint was mirrored by that of Participant 4, who said:


*“I can tell you most of my appointments are about taking new medicine supply. Every time, the doctor asks you by saying quickly, how are you? Are you good? And write the prescription without leaving any chance to discuss my condition. I am telling you, the medical visit is only about taking new medicine supply”*
(Participant 4).

Participant 4 sensed from the beginning that his physician was not interested in initiating a conversation but was more interested in ending the consultation as soon as possible. This dismissiveness was evident in the physician’s closed-ended questions. This sub-theme illustrates that the majority of the patients viewed their medical experiences as disappointing because they felt that physicians were not paying them enough attention. In contrast, a number of participants provided accounts of medical consultations where they experienced a feeling of being noticed and respected. Lena described her positive experience: when I went to Riyadh, I met a doctor who answered all of my questions and informed me about my health case. Clearly, a physician’s interpersonal skills are central to patient concern. The patients’ descriptions suggest that the physicians’ relational skills are as important as their medical skills. This is represented by the following quote:


*“I think some visits are happy because once the doctor starts to talk to you and listen to you then as a patient you feel mentally good. I think half of the treatment happens when the doctor start to exchange information with you”*
(Participant 4).

Participant 4 classified his medical encounters as either happy or unhappy.


*“A happy encounter occurs when meeting a physician with good interpersonal skills which enables effective interaction. Patients come to see physicians with two things in mind. First, to be cured, and second, to feel better by knowing what is wrong with them”.*


This was described by Participant 4 as only half of the treatment. He believes that half of the medical treatment can be accomplished through communication, which he referred to as information exchange. Similarly, Participant 7 describes her physician being ‘nice’ to her and taking time to listen to her concerns.


*“He was so nice with me because he listens to me and answers all of my questions. He explained to me exactly what my illness is and how the treatment is going to be. We exchanged information. He showed me his attention”*
(Participant 7).

She was satisfied because she met a physician who appeared to have good interpersonal skills that enabled her to connect and communicate with him. From the patients’ perspective, doctors can show reciprocity by listening to patients who are eager to tell their story and reveal their concerns and fears, and in return, explain any concerns and eliminate any uncertainty. The importance of the exchange and reciprocity during their consultations was mentioned repeatedly by patients, in defining what they considered as effective communication. The exchange of information was used by patients as a measure to evaluate their communication experiences. This sub-theme showed that the majority of the patients’ medical experiences were disappointing because physicians were not paying enough attention to them, which resulted in the inability to form a reciprocal partnership, which resulted in the patients’ reluctance to trust non-Saudi physicians, as the next sub-theme will illustrate.

#### 3.2.3. Sub-Theme Three: Reluctance to Trust

The patients’ reluctance to trust featured strongly in many of their descriptions of their medical consultations. Their inability to trust reflected negatively on the formation of physician-patient partnerships. The basis of this lack of trust lay in the patients’ views of the physicians’ professional competence, and in the patients’ belief that non-Saudi physicians are motivated by money and are not highly qualified. Patients went to see physicians to be cured of an ailment, or at least to have an accurate diagnosis. Participant 6 stated: patients are looking to get accurate diagnoses more than anything else. Participant 2’sexperience highlights how a perception of incompetence can result in a loss of trust:


*“After the ambulance brought me to the hospital, the doctor took some quick examination on me and hospitalized me. He insisted on me to walk little from my bed to the bathroom in the same room. I did once and it was horrible. Later on, the doctor run CT scan on me. The scan shows that I have multiple fractures in three parts in my body. The doctor came to me and apologized for forcing me to walk when I was not ready for that. I lost trust on them because of their inaccurate diagnosis and their careless behavior. I wonder why the doctor did not do a ST scan on me from the beginning”*
(Participant 2).

This physician ignored Participant 2 as a valuable source of information in relation to her own experience. While the physician made his decision that she was able to walk based on his personal opinion supported by a quick examination, he rejected her opinion, which was based on her feeling that her body was telling her not to attempt to walk. She went through a terrible experience where she felt that the physician did not act in her best interest. Hence, she lost her trust in the physician, which might affect her future cooperation with other physicians. Additionally, she felt that her safety was put at risk because the CT scan showed that she had multiple fractures and forcing her to walk could have resulted in further fractures. Participant 6′s medical experience confirms the importance of the physician’s medical competence in building and sustaining patient trust. Participant 6 stated:


*“I explain my knee problem briefly to the non-Saudi physician. He asked me to get in the bed, then he conducted a hand examination on my injured knee. He diagnosed me as having torn ligaments and suggested a surgery after meeting him. I asked him to make X-ray to me. He refused and told me that he knows better than me. Here, doctors do not 110 want to hear your opinion or to have a say over anything. It is funny because after all it is your body and health. However, I insisted on making X-ray before I go into any type of surgery. So, he prescribed X-ray to me. The result showed a minor torn ligament, so I decided not on the surgery but on the physical therapy”*
(Participant 6).

Participant 6 felt that his physician was not competent enough because he decided on surgery based only on a palpation examination. He expected the physician to at least order an X-ray without having to insist on it. When the result of the X-ray showed a minor torn ligament, Participant 6 decided on a therapist treatment instead of a surgical procedure. There is a strong sense of anger and disappointment in Participant 6’s story because the physician ignored his view and excluded him in the decision-making. By refusing his request for an X-ray, the physician failed, not only to diagnose the injury, but also to engage Participant 6 in the treatment process. Hence, Participant 6 was dissatisfied with the outcome and may in future be reluctant to trust physicians. Participant 5 also expressed concerns about trusting physicians:


*“For me, I don’t trust all doctors regardless of nationality because I have seen three doctors in three hospitals regarding my health conditions and all of them have given me different diagnoses. You see now why I don’t trust them. One doctor asked me to get exposed to the sun light as a treatment while the other doctor has required a laser treatment along with a skin lotion only. Now, I stopped the sun treatment and doing the laser but with no improvements”*
(Participant 5).

Unlike Participants 2 and 6, Participant 5 indicated that he had lost trust in physicians due to his perception of their medical incompetence, regardless of nationality. There is a strong sense of suffering in Participant 5’s stories because he does not yet know the root of his health problem. There is also disappointment attributable to three inaccurate diagnoses so far. As a result, Participant 5 not only lost trust in physicians, but also his adherence to the medical regime was impacted because he chose to stop the sun treatment.

However, Participant 8’s medical experience as described below was positive. His experience illustrates how gaining a patient’s trust can result in patient engagement and satisfaction. Participant 8 had no hesitation in trusting the physician because he experienced good communication and a physician who had the medical competence and skill to diagnose the illness:


*“One of the doctors I met was non-Arabic. This doctor gains my trust because he asked me couple of questions just to get exactly the type of dizziness I have. His questions were like that do you feel dizzy just before stand up, after standing up completely, or do you feel dizzy after setting to long and then standing up”*
(Participant 8).

So far, the participants’ examples have demonstrated how a physician’s medical skills can jeopardize or gain the patient’s trust. However, patients’ experiences showed that a lack of trust toward non-Saudi physicians was also attributable to preconceptions and stereotyping of non-Saudi physicians. Some patients believe that non-Saudi physicians are not qualified and are only employed for financial money purposes as the next quote explicitly illustrates:


*“Saudi patients have less trust toward non-Saudi doctor. Some believe that they are not good enough and other believe that non-Saudi doctors are driven only by money incentives”*
(Participant 5).

However, patients’ views differed regarding trust. Participant 6 adopted the same position as Participant 5, that physicians are motivated by money: some doctors are motivated by money to serve the patients. Participant 6 agreed with Participant 5 concerning qualifications and stated: I believe that some doctors are not qualified enough to work here. Some of them are not good to the point that it makes me wonder if they actually went to a medical college. Participant 8drew a comparison:


*“Most Saudi doctors are qualified and skillful and able to gain the patient’s trust. I think, you will be lucky if you can get treated by Saudi doctors because they are better than non-Saudi doctors”*
(Participant 8).

Participant 8’sperspective indicates a preference for Saudi physicians because non-Saudi physicians are presumed to be less skilled and qualified than Saudi physicians, so Saudi patients are reluctant to trust non-Saudi physicians. This sub-theme illustrates the patients’ lack of trust in non-Saudi physicians. This was attributed to a perception of medical incompetence, and to the belief that expatriate physicians are primarily motivated by money and are not well qualified. The lack of trust has influenced the patients’ engagement, adherence to treatments, their cooperation in future medical encounters, and, most importantly, prevented and discouraged them from establishing a rapport and partnership with their physician. However, as the next sub-theme illustrates, the amount of time set aside for consultations, and the way that time was used by physicians, also influenced the possibility of an effective physician-patient partnership.

#### 3.2.4. Sub-Theme Four: Not Being Given Adequate Time

The fundamental cause of the patients’ inability to establish a rapport and form a partnership with their physicians was attributed to insufficient time given by their physicians. This implies, from the patient’s experiences, that they perceived that physicians are in charge of allocating the timeframe for consultations. This was reflected in the patients’ experiences in which they expressed the notion that they were a burden and unworthy of the physician’s time, when in fact their rights and privacy had been violated, and their needs and expectations unmet. Some patients felt that physicians provided care for them as a favour and not as a professional obligation. Participant 2 described a display of what she perceived to be humiliation inflicted on patients by patronizing physicians during the medical encounter:


*“Some doctors try to show you that they are so busy and you as a particular patient are an extra burden over his daily load. So, they expect you to understand this message. I believe that the main reason of being a doctor is to serve patients because that is the doctors’ duty. They should understand that when they serve patients, they are doing no favour to anyone but it is their duty”*
(Participant 2).

From Participant 2’sperspective, physicians often attempt to minimize the time allocated to patients. She believes that physicians often treat patients in a condescending manner that makes the patients feel that the physician is doing them a favour; that physicians are frequently dismissive of patients during the medical encounter. There is a strong sense of anger and disgust for the physician’s lack of compassion, irrespective of the fact that he is failing to fulfil the primary function of his profession, to attend to patients. Participant, 6 also expressed his outrage at physicians who he perceives are not willing to commit time to attend appropriately to patients’ needs and who behave as if they own the healthcare facility:


*“They should understand that I didn’t come to see them in their houses but in the public hospital. The above examples indicate that some physicians refrain from dedicating enough time for patients which results in patients feeling they are imposing on the physicians’ time”*
(Participant 6).

Thus, patients feel disempowered and belittled in attempting to develop relationships and build partnerships with their physicians. It takes time to establish a relationship, and the time given to physicians for consulting appears to be insufficient in outpatient medical encounters. Participant 2 made the following comparison: 

“*Doctors talk and listen to us more when I was hospitalized in the hospital because they have more time*”. In the patient’s quote below, Participant 6 describes experiencing a connection and mutual understanding with her physician after she had surgery and was hospitalized. Her experience implies that patients are not allocated enough time in outpatient medical encounters.


*“He had run my surgery with a success thanks for god. Now after spending so much time with him, we have good relationship. He understands me now better and knows what I like and what I do not like”*
(Participant 6).

All patients are the same in the sense that they have questions and concerns and arrive with the expectation that these will be dealt with and heard by someone with the knowledge to provide answers. However, if the physician is not willing to dedicate time, then the patients’ expectations will not be met, as Participant 1 confirms:


*“The medical visit was too short, I did not get enough chances to ask all of my questions that are in my mind. You know that when we go to a doctor, we all have concerns and questions that need to be answered from a person who knows”*
(Participant 1).

Similarly, Participant 5 blames the physician for not being informative. He felt that the physician was more concerned about the time spent rather than the patient.


*“My problem comes when doctors don’t give me details about my health problem because they don’t have enough time and want to serve the next patient”*
(Participant 5).

The time spent with physicians was seen as a critical element that impacted on patients’ ability to establish a connection, and a mutual understanding. Participant 3 stated: 


*“The medical visit time does not help either patient or doctor to understand each other because it is short. These experiences confirm that insufficient time allocated physicians to consult with patients was a barrier to meeting patients’ needs and expectations, to ensuring patient satisfaction, and to establishing patient-centered care”.*


Participant 6 described a situation in which his right to privacy was compromised and where there was a blatant violation of his rights.


*“However, he did not listen to me enough and he was kind of pushing me to cut my talk. During the last part of the medical visit, he allowed another patient to sit in the chair facing me. I objected right away but the doctor said it is ok because there is a lot of patients waiting outside and your visit is about to end. I insisted that it is still my time and the other patient must leave the room. Finally, the doctor said that I am right and asked the other patient to leave the room, which he did. However, I felt that the doctor is somehow not happy with me because of my stand on this issue”*
(Participant 6).

Participant 6’s experience shows just how much pressure patients experience by not being given adequate time. He was not only rushed to finish his medical visit but also his personal rights and the ethics of medical practice were violated when the physician invited the next patient to enter the room. When he insisted that the other patient must leave the room, he sensed that his objection was not welcomed by his physician because it was against what the physician wanted. Participant 6 saw his experience as a violation of the physicians’ professional oath that requires all physicians to act in the best interest of patients, as well as a contradiction of the ethical principles that guarantee the privacy and confidentiality of patients. Participant 2 recounted a similar experience:


*“Sometime, I get surprised because another patient is given the permission to enter the room while my visit still not yet finished. I protested against that. It is funny because there was a sign in the doctor’s door that says “it is not allowed for a patient to enter the room if there is another patient still inside”*
(Participant 2)

Participant 2 was surprised that the physician could allow another patient access to the room while her medical consultation was still in progress. For her, this was unacceptable and in obvious conflict with the hospital’s policy and universal human right to patient privacy and confidentiality, which it was disgraceful for the physician to ignore. To conclude, this sub-theme has illustrated how the consultation time allocated to participants was inadequate to meet their needs and expectations. A satisfactory consultation is one that ensures exclusive time and space for each patient to meet with their physician to discuss their health concerns. Thus, the patients’ medical experiences resulted in dissatisfaction and, on occasions, violations of their rights.

## 4. Discussion

It is well documented that successful medical consultations depend on effective communication characterized by the sharing of information between patients and doctors in which the meaning is mutually understood [22,23,24]. Empirical research that has examined intercultural medical encounters has demonstrated that cultural differences between physicians and patients have a significant impact on the quality of communication and the provision of care. However, most of these studies only emphasized differences in language. This present study shows clearly that patients come to the medical consultation with not only health problems to be treated, but also with strong attachments to a set of values and beliefs. These findings are consistent with previous research that highlights distrust as a major issue in intercultural encounters. Studies have identified that distrust poses barriers toward establishing the physician-patient relationship, good rapport, and partnership, and prevents the patients’ disclosure of health information, and patient medical compliance [25].With regard to trust and its association with the patient’s negative perception of the medical competence of international physicians, our findings have shown concordance with most of the studies in the literature, including studies conducted in Saudi Arabia [10,23,26]. However, some studies conducted in Saudi Arabia have shown patients displaying a positive perception of the medical competence of international physicians [27,28]. Effective communication refers to the ability of physicians and patients to exchange information with the meaning mutually understood [29]. This study indicates that the language difference between Saudi patients and non-Saudi physicians has led to ineffective communication, the patients’ dissatisfaction, the physicians’ frustration, and other health outcomes, such as misdiagnosis and compromises on the patients’ health safety. These findings are congruent with several studies conducted in Saudi Arabia and other countries that linked the absence of common language to the patients’ dissatisfaction, misunderstandings, misdiagnoses of health problems, and tense relationships [22,24,30]. For Bourdieu, habitus shapes and makes up people’s identity that includes, but is not limited to, all of the dispositions of custom, values, religion, and even health belief. In intercultural encounters, these differences greatly constrain the ability of physicians and patients to construct a relationship with effective communication, and this was a barrier toward the patients’ satisfaction, physician performance and job satisfaction, provision of optimal healthcare, and important clinical outcomes, such as patient health safety and patient adherence to the prescribed treatment. Our findings are consistent with other studies [23,31].

Bourdieu extended the concept of capital beyond the conventional economic application to anything that adds value or power to its holders and provides access to particular fields. In this study, the main constraint that affected the physician-patient interaction was the absence of ‘linguistic capital’ [32]. Not being able to speak the patients’ language disempowered non-Saudi physicians on a relational, communicational, and professional level. Non-Saudi physicians experienced numerous constraints due to encountering Saudi customs, traditions, and religion that were unknown and different from their own. They lacked the cultural codes to greet a person and start a conversation, and how to treat the elderly and female patients in particular. They were lacking the ‘cultural capital’ that assigns position, power, and entry for its holders in the field of the medical consultation [33]. In life, people would use cultural knowledge as capital to strengthen their position, to establish a conversation and maintain connections with others. The lack of this type of capital in the non-Saudi physicians had undermined their ability to interact with Saudi patients effectively. Similar to physicians, the patients’ little medical knowledge meant that they lacked capital that would have positively influenced their interaction with physicians. The patients’ limited medical knowledge and understanding of medical terminology constrained their ability to engage and participate in decision-making. Thus, our findings demonstrate that insufficient time for the medical consultation coupled with substantial workload had led to ineffective communication, physician job dissatisfaction, the patients’ dissatisfaction, and the provision of suboptimal care. Applying Bourdieu’s theory of habitus, capital, and field to understanding of physician and patient behavior helps in determining the likelihood of engaging in effective communication.The analysis of the patients’ experiences of their medical encounters revealed that the effectiveness of physician-patient communication was jeopardized by two challenges. The first challenge is illustrated by the essential role that difference plays in shaping the patient’s experience. The patients’ perception of difference was preconceived before meeting their physicians and this negatively influenced their apprehension with regard to effective communication. Moreover, the disregard for cultural sensitivities and differences by physicians intensified the degree of the patients’ perception of difference, causing communication disconnection between patients and physicians which eventually led to patient dissatisfaction, disillusionment, and in some cases to the patients’ safety being compromised. However, the patients’ perception of difference was mitigated whenever their cultural values and needs were acknowledged and accommodated by physicians.

The second challenge is the absence of partnering in the patient-physician relationship that had a negative impact on the patients’ ability to establish a rapport with their physicians and prevented a partnership formation. Patients felt powerless, overlooked, unwilling to trust non-Saudi physicians, and perceived that the time allocated to their consultations was inadequate to meet their needs and expectations. As a consequence, the patients’ medical experiences resulted in ineffective communication, dissatisfaction, and ethical violations of patient rights. These factors, together with the resultant lack of trust, compromised patient safety because it resulted in non-adherence to medical regimes. This chapter has analyzed the physician-patient communication phenomena from the patients’ perspective.

## 5. Conclusions

To our knowledge, this is the first qualitative study to investigate the communication experiences of Saudi patients and non-Saudi physicians during medical consultations. This study represents a step toward improving the understanding of the nature of the phenomena of intercultural communication in medical consultations, particularly in Saudi Arabia. The study findings highlight that physician-patient communication in this context is much bigger than what is happening in the room at the time. It is influenced largely by cultural practices and concerns, as well as systems that operate within healthcare organizations. Moreover, physician-patient communication is greatly impacted by preconceived perceptions of difference. Differences, whether preconceived, constructed, reinforced, or mitigated, are at all times front and center in the consultation experience. Difference, and its impact, was found to be the overarching theme of the medical interaction between non-Saudi physicians and Saudi patients. Preconceptions, together with the actual encountering of difference set in the motion process of skepticism, distrust, and othering, a situation in which the verbalization of concerns and the sharing of critically important information does not occur. Without the urgent attention ofall stakeholders, the situation will not improve. Optimal outcomes for patients and physicians can only be achieved by engaging stakeholders in strategic practical change, guided by sound knowledge of what is important and what is possible for all stakeholders.

## 6. Implications for National Policy

Some practical improvements could be implemented by the Ministry of Health to support organizations to fund physician-patient communication. The Ministry of Health could consider instituting an independent department within its Ministry, whose purpose is solely dedicated to the planning and delivery of healthcare that cannot be compromised by the cultural and linguistic diversity of its physicians. This department’s main objective would be to ensure that the health services delivered are appropriate formeeting the patients’ cultural and linguistic needs and the department would be proactive in collaborating with all stakeholders, namely organizations, health workers, and patients, in constructing guidelines and recommendations that promote patient-centered, culturally appropriate healthcare. More importantly, this department should exercise its authority as part of MOH to aid health organizations in the establishment of governance systems to maintain and sustain culturally appropriate patient centered health care. Additionally, at a national level, the Saudi Arabian government could put more pressure on medical schools to enhance their current medical curricula, particularly in relation to communication. Medical schools are urgently advised to revise curricula that are tailored mainly to an evidence-based approach to healthcare communication. Saudi medical schools have been unsuccessful in supplying the necessary quota of medical graduates to meet the health needs of the Saudi population. Our study indicates that increasing the proportion of Saudi physicians would facilitate the delivery of culturally appropriate holistic care. Thus, close cooperation between the Ministry of Education and the Ministry of Health is required in order to reduce the gap between the demand and supply of physicians, including female physicians. The Ministry of Education is advised to initiate and implement strategies to attract more female physician candidates by considering culturally appropriate ways to deliver medical education that is tailored to suite the conservative nature of the Saudi society and facilitate women participating in medical education without having to compromise their cultural practices. The Ministry of Health is encouraged to provide a suitable work environment that takes into consideration the cultural aspects of the Saudi society to attract more women into the medical field. Increasing the number of female Saudi physicians in the workplace will enhance the delivery of care to women.

## 7. Implications for Clinical Practice and Education

Findings of this study instruct hospital administrators to take precautionary steps to mitigate the negative impact caused by the differences between non-Saudi physicians and Saudi patients. Improving the expatriate physicians’ cultural competence requires educating them about Saudi culture and training them to apply such knowledge in practice. This can be accomplished through running workshops or short courses on how to provide cultural congruent care. Another self-teaching method that could be more suitable, considering the physicians’ busy schedules, is the utilization of “Interactive multimedia training” containing educational materials that could be accessed on CD/DVDs, smartphones, or hospital websites [34,35]. In addition, on the job strategies can be implemented to reduce the impact of cultural differences as a negative influence.

## 8. Limitations and Recommendations for Future Research

This study was conducted in one site only and hence findings must be tested in application to other sites in other parts of Saudi Arabia and in other similar countries. A further limitation of this study was the cultural barrier that was encountered by the researcher during the interview process [35]. This may have limited the researcher’s full understanding of the female patients’ medical experience in person, such as in facial expression and body language. This study has revealed the scarcity of research in the area of physician-patient intercultural communication within the context of Saudi Arabia, together with the need for more studies to examine in depth ways in which communication can be improved. Therefore, future studies on this aspect may involve intercultural mediators during the interview process. Even though previous studies have identified communication problems between non-Saudi physicians and Saudi patients [36], this study is the first study to examine communication from both the physicians’ and patients’ perspective. As there are only eight informants in this study, which influences the consistency of the result, further qualitative research is recommended involving more non-Saudi physicians and Saudi patients, both male and female, to better understand patients, and particularly the womens’ experiences and ways in which their healthcare outcomes can be improved. Similarly, the impact of the patients’ perception on their own behavior and interaction with physicians needs to be further understood. Future studies can take advantage of our findings by applying and testing the current study’s conceptual framework. With the advent of the COVID-19 epidemic and the consequent increase in tele-health medical consultations, and the likelihood of its continued use in the future, urgent research is required to understand how intercultural communication occurs and may be compromised or enhanced in this context, particularly in Saudi Arabia. There is still a place for improvement and further discoveries with regard to the issue under study, especially in Saudi Arabia where there exists a scarcity of empirical research investigating physician-patient communication. Further research is required that examines the communication of both Saudi physicians and non-Saudi physicians, and the ways in which communication and consequent health outcomes can be improved.

## Figures and Tables

**Table 1 healthcare-11-00118-t001:** The inclusion and exclusion criteria of the study participants is as follows.

PICOS	Inclusion Criteria	Exclusion Criteria
Population	All Physicians (consultants, specialist, GPs) and patients come from different cultural backgrounds	Physicians and patients who share the same culture communication with health providers or health staff other than physicians
Intervention	Communication during medical consultation	Inpatient medical consultation, or any medical consultation in nursing homes and/or homebased care
Comparison	None	None
Setting	Out-patient clinics and other clinics	Hospitalized patients, including mental health patients
Outcome	Factors hindering communications	-

**Table 2 healthcare-11-00118-t002:** Demographic Characteristics of the Participants.

Participant	Gender	Age	Education	Job	Nationality
Participant 1	Male	35	Bachelor	Teacher High school	Saudi
Participant 2	Male	59	High school	Govt Worker	Saudi
Participant 3	Male	28	College student	Qassim University	Saudi
Participant 4	Male	29	Bachelor	Govt Civil Employee	Saudi
Participant 5	Female	24	College student	Qassim University	Saudi
Participant 6	Female	22	College student	College of Technology	Saudi
Participant 7	Female	40	High school	Housewife	Saudi
Participant 8	Female	24	Bachelor English	Qassim University	Saudi

**Table 3 healthcare-11-00118-t003:** Main Themes and Sub-Themes.

S#	Main Themes	Sub-Themes	Units of Meaning
1	Encountering Difference	Being different	Involve talking with other backgrounds? Expectation Personal experience Fulfillment experience Good or bad experiences? Outcome Patient satisfaction? Suggestions regarding cross cultural communication between doctors and patients?
		Being disconnected	
		Being culturally acknowledged	
22	Absence of Partnering	Being powerless	
		Being unnoticed and neglected	
		Reluctance to trust	
		Not being given the adequate time	

## Data Availability

Data is available on request from Mohammad Alkhamees.

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
