# Peer review of "A Qualitative Investigation of Factors Affecting Saudi Patients’ Communication Experience with Non-Saudi Physicians in Saudi Arabia"

_healthcare, 2022, doi:10.3390/healthcare11010118_

Round 1

Reviewer 1 Report

Dear Authors,

Thank you for the opportunity to review this article, on such a pertinent topic in the health area. The article presents a wealth of information, however, some aspects should be improved, which I will now present.

Abstract:

Please remove the name of the Hospital.

Rather than referring to the study population, you should refer to the study participants.

 Keywords:

Please remove this keyword: King Fahad Specialist Hos-41 pital.

I suggest including the following keywords: Communication; culture.

Introduction

Please review the objective. This study only explores the perception of patients.

Remove results presents in the Introduction.

Theoretical support in the Introduction is very poor, please make it richer

Materials and Methods

Please rearrange the information you have on this item.

2.1.1. Sampling

Don't talk about sample. This is not a quantitative study. Define a subtitle - Participants.

Include here the inclusion and exclusion criteria of study participants.

Suggestion: Transfer the Table and the analysis on the characteristics of the participants to the Results.

2.1.2. Data Collection Methods 

Please replace and place only: Data collection

2.1.3. Data Analysis Techniques

Please give more details about the thematic analysis process. Mention why you only did 8 interviews

2.1.3 Ethical Considerations (Please include the subtitle)

Transfer some aspects that you have in the previous subpoints to here

What ethical considerations were followed

Which Ethics Committee approved this study?

What is the Approval number?

Please remove the name of the Hospital.

Throughout the analysis, you use participant names. Please remove and use for example letters. Please mention how you protected the identity of the participants

Results

Remove names from what the participants mentioned. Always put in a different font what the participants said. Line 137 and following, the letter is not different from the one used in your analysis.

You have interesting results, but the text is too long. You cannot select the most significant aspects.

Discussion and Conclusions

It is very poor, mobilize more scientific evidence for the Discussion.

Author Response

To

The Editor

Healthcare MDPI

Subject: Submission of Revised article

Respected Editor

It is stated that we are thankful for sending us the valuable comments. We have tried our best to incorporate the comments/suggestions given by respectable reviewers. We have answered almost all comments and provided justification as well. The incorporated new information in red color represents Reviewer 1 and yellow color reviewer 2. If any text is highlighted in red and yellow both color it represents the same comment from both reviewers.

Regards

Dr. Yasir Hayat Mughal and co-authors

Response to reviewer 1

S#

Reviewer 1 Comments/Suggestions

Authors’ Response

1

Abstract

1.Please remove the name of the Hospital.

2. Rather than referring to the study population, you should refer to the study participants.

Thank you very much for the valuable comments following are the authors’ action on the reviewer’s comments:

1. Name of the hospital is removed from abstract as suggested by the reviewer.

2. Study population is replaced by study participants in the abstract.

2

Keywords:

1.       Please remove this keyword: King Fahad Specialist Hospital.

2.       I suggest including the following keywords: Communication; culture.

1.    Name of King Fahad Specialist Hospital is removed from keywords.

2.    Following words suggested by reviewer has been added (Communication; culture).

3

Introduction

1.       Please review the objective. This study only explores the perception of patients.

2.       Remove results presents in the Introduction.

3.       Theoretical support in the Introduction is very poor, please make it richer

1. This objective is reviewed

2. Results presented in the introduction are removed.

3. Strong theoretical work from social and cultural reproduction theory has been added in detail in the introduction section as suggested by the reviewer.

4

Please rearrange the information you have on this item.

2.1.1. Sampling

Don't talk about sample. This is not a quantitative study. Define a subtitle - Participants.

Include here the inclusion and exclusion criteria of study participants.

Suggestion: Transfer the Table and the analysis on the characteristics of the participants to the Results.

1.       A subtitle participant is created.

2.       An inclusion and exclusion criterion of participants is provided in the table form.

3.       Characteristics of the participants Table is transferred to results sections

5

2.1.2. Data Collection Methods 

Please replace and place only: Data collection

As suggested by the respected reviewer only Data Collection is placed and highlighted in red.

6

2.1.3. Data Analysis Techniques

Please give more details about the thematic analysis process. Mention why you only did 8 interviews

Thematic analysis is discussed in more detail under data analysis techniques.

According to Clarke (2010) and Noon (2018) 4-10 interviews are enough for doctoral studies. As this paper is part of PhD thesis that is why researcher has conducted eight interviews which fulfil the requirement. Second length of interviews was long enough researcher was able to extract sufficient information from eight interviews.

7

2.1.3 Ethical Considerations (Please include the subtitle)

Transfer some aspects that you have in the previous subpoints to here

What ethical considerations were followed

Which Ethics Committee approved this study?

What is the Approval number?

Please remove the name of the Hospital.

Throughout the analysis, you use participant names. Please remove and use for example letters. Please mention how you protected the identity of the participants

Some subpoints are transferred in ethical consideration section as suggested, remaining detailed information about ethical consideration, ethics committee approval number are given in the section 2.1.3.1

Hospital name is removed from previous sections.

Names of the participants are removed.

Protection of the Identity of the study participants is also explained in section 2.1.3.1

8

Results

Remove names from what the participants mentioned. Always put in a different font what the participants said. Line 137 and following, the letter is not different from the one used in your analysis.

You have interesting results, but the text is too long. You cannot select the most significant aspects.

Authors tried their best to remove All the names of the respondents and all participants’ words are now italic.

Thanks for the comment respected reviewer, as the interviews last for one hour and more and qualitative study must be rich with lot of in depth information that is the reason the length of the text is long. 

9

Discussion and Conclusions

It is very poor, mobilize more scientific evidence for the Discussion.

As suggested by the reviewer authors have tried their best to enrich discussion section and compare with past studies with more scientific evidences.

Reviewer 2 Report

The subject of the manuscript is interesting as barriers in health communication determine the therapeutic relationship.

In the abstract, the phrase "female interviews were necessarily conducted by a female interviewer" should be expressed differently, as interviews between men and women do not have to be censored, even if the study is conducted in Saudi Arabia, nor should it appear in point 7.

The introduction manages to bring us closer to the importance of health communication, although it is too brief and with too few bibliographical sources. I miss a paragraph describing what the health system in Saudi Arabia is like and some data on foreign health professionals; percentage of the total, nationalities, etc. in order to better understand the scale of the problem. In the last paragraph after the objective of the study is presented and the results are discussed, all of this should be deleted. The introduction is not the abstract.

The use of qualitative methodology is very opportune but it is necessary to be more rigorous and orderly in its description. I recommend the following order; methodology (type of qualitative research), description of the research context (where it took place) and description of participants with inclusion and exclusion criteria, data collection technique (interview structure and key questions/themes addressed), why only 8 interviews, was it a pre-determined number or was there data saturation, data analysis and ethical considerations.

For the presentation of the results it helps to include a table with themes, sub-themes and units of meaning. It is also important to identify the participant in the verbatim. In my opinion, the verbatim used are too long and the results are too extensive; they should be reduced and the explanations of themes and sub-themes should be more specific.

The discussion has to "discuss" the results obtained with other similar studies, in this manuscript it is not done. It is mixed with the conclusions and neither the discussion nor the conclusions are clear. A real discussion should be made and then clear and concise conclusions should be drawn.

In the final sections I suggest reflecting on the possibility of having intercultural mediators to correct the problems detected.

The limitations of the study should reflect the fact that there are only 8 informants, which influences the consistency of the results.

Author Response

To

The Editor

Healthcare MDPI

Subject: Submission of Revised article

Respected Editor

It is stated that we are thankful for sending us the valuable comments. We have tried our best to incorporate the comments/suggestions given by respectable reviewers. We have answered almost all comments and provided justification as well. The incorporated new information in red color represents Reviewer 1 and yellow color reviewer 2. If any text is highlighted in red and yellow both color it represents the same comment from both reviewers.

Regards

Dr. Yasir Hayat Mughal and co-authors

Response to Reviewer 2

The subject of the manuscript is interesting as barriers in health communication determine the therapeutic relationship.

Thank you very much for the valuable comments following are the authors’ action on the reviewer’s comments:

In the abstract, the phrase "female interviews were necessarily conducted by a female interviewer" should be expressed differently, as interviews between men and women do not have to be censored, even if the study is conducted in Saudi Arabia, nor should it appear in point 7.

Accepted the reviewer’s comments authors feel that this sentence in the abstract may not be required.

The introduction manages to bring us closer to the importance of health communication, although it is too brief and with too few bibliographical sources. I miss a paragraph describing what the health system in Saudi Arabia is like and some data on foreign health professionals; percentage of the total, nationalities, etc. in order to better understand the scale of the problem. In the last paragraph after the objective of the study is presented and the results are discussed, all of this should be deleted. The introduction is not the abstract.

Information about Saudi health system, some data on foreign and local health professionals have been added in the introduction and highlighted in yellow color. Moreover results after the objective of the study in the introduction sections are removed.

The use of qualitative methodology is very opportune but it is necessary to be more rigorous and orderly in its description. I recommend the following order; methodology (type of qualitative research), description of the research context (where it took place) and description of participants with inclusion and exclusion criteria, data collection technique (interview structure and key questions/themes addressed), why only 8 interviews, was it a pre-determined number or was there data saturation, data analysis and ethical considerations.

Thanks for the comment. As per the suggestions of the respected reviewer the order is followed such as methodology/study design, description of the research context, inclusion and exclusion criteria of the study participants, data collected technique, main interview question with main themes.

According to Clarke (2010) and Noon (2018) 4-10 interviews are enough for doctoral studies. As this paper is part of PhD thesis that is why researcher has conducted eight interviews which fulfils the requirement. Second length of interviews was long enough researcher was able to extract sufficient information from eight interviews.

For the presentation of the results it helps to include a table with themes, sub-themes and units of meaning. It is also important to identify the participant in the verbatim. In my opinion, the verbatim used are too long and the results are too extensive; they should be reduced and the explanations of themes and sub-themes should be more specific.

Thanks for suggestion, authors have created a table in which main themes and sub-themes are added .

Thanks for the comment respected reviewer, as the interviews last for one hour and more and qualitative study must be rich with lot of in depth information that is the reason the length of the text is long. 

The discussion has to "discuss" the results obtained with other similar studies, in this manuscript it is not done. It is mixed with the conclusions and neither the discussion nor the conclusions are clear. A real discussion should be made and then clear and concise conclusions should be drawn.

` As suggested by the reviewer authors have tried their best to enrich discussion section and compare with past studies with more scientific evidences.

Separate and concise conclusions have been drawn as suggested by the reviewer.

In the final sections I suggest reflecting on the possibility of having intercultural mediators to correct the problems detected.

The suggested intercultural mediators are added in the last section and highlighted in yellow color.

The limitations of the study should reflect the fact that there are only 8 informants, which influences the consistency of the results.

This limitation is addressed in the last section and highlighted in yellow.

Round 2

Reviewer 2 Report

I would like to congratulate the authors of the manuscript for the modifications introduced. The paper has improved a lot, it is publishable in a quality journal like IJERPH.